# Fecal Microbial Transplantation in Critically Ill Patients—Structured Review and Perspectives

**DOI:** 10.3390/biom11101459

**Published:** 2021-10-04

**Authors:** Ivana Cibulková, Veronika Řehořová, Jan Hajer, František Duška

**Affiliations:** 1Third Faculty of Medicine, Charles University, 11000 Prague, Czech Republic; ivana.cibulkova@fnkv.cz (I.C.); veronika.rehorova@lf3.cuni.cz (V.Ř.); jan.hajer@fnkv.cz (J.H.); 2Department of Medicine, FNKV University Hospital, 10034 Prague, Czech Republic; 3Department of Anesthesiology and Intensive Care Medicine, FNKV University Hospital, 10034 Prague, Czech Republic

**Keywords:** gut microbiota, critically ill, fecal microbial transplantation, multiorgan failure

## Abstract

The human gut microbiota consists of bacteria, archaea, fungi, and viruses. It is a dynamic ecosystem shaped by several factors that play an essential role in both healthy and diseased states of humans. A disturbance of the gut microbiota, also termed “dysbiosis”, is associated with increased host susceptibility to a range of diseases. Because of splanchnic ischemia, exposure to antibiotics, and/or the underlying disease, critically ill patients loose 90% of the commensal organisms in their gut within hours after the insult. This is followed by a rapid overgrowth of potentially pathogenic and pro-inflammatory bacteria that alter metabolic, immune, and even neurocognitive functions and that turn the gut into the driver of systemic inflammation and multiorgan failure. Indeed, restoring healthy microbiota by means of fecal microbiota transplantation (FMT) in the critically ill is an attractive and plausible concept in intensive care. Nonetheless, available data from controlled studies are limited to probiotics and FMT for severe *C. difficile* infection or severe inflammatory bowel disease. Case series and observational trials have generated hypotheses that FMT might be feasible and safe in immunocompromised patients, refractory sepsis, or severe antibiotic-associated diarrhea in ICU. There is a burning need to test these hypotheses in randomized controlled trials powered for the determination of patient-centered outcomes.

## 1. Introduction—Defining Human Gut Microbiome

The term microbiota refers to the community of microorganisms (comprising of bacteria, archaea, fungi, protozoa, and viruses) that inhabit a particular environment. Growing attention is attributed to the microbial communities associated with various niches in the human body. Their genomes (genes and plasmids) are referred to as the microbiome. It is estimated that the microbiota of a healthy human consists of between 500 and 2000 species [1]. The density of microorganisms is highest in the colon and the gross majority of bacteria are strict anaerobes [2]. The gut microbiota is indispensable for a range of aspects of healthy human physiology. Most notably, microbiota influence gastrointestinal motility, regulate mucosal barrier function and epithelial cell turnover, influence immune responses, and suppress pathogen overgrowth. Indeed, they also play an important role in the host metabolism, converting dietary fiber to short chain fatty acids (SCFA), which serve as energy substrate for colonocytes. Butyrate producers are also protective against mucosal inflammation and infection [3].

## 2. Intestinal Microbiota Diversity and Relation to Immunity and Inflammation

The gut microbiota is a dynamic ecosystem shaped throughout human lifespan, from prenatal conditions (mother’s health and fetus’s genetic factors), mode of birth (Caesarean section versus vaginal delivery), diet, BMI, weight, environment, and antibiotic exposure to hospitalizations during later life. The gut microbiota of the adults is dominated by taxa belonging to two phyla, Bacteroidetes and Firmicutes, with their relative proportions differing among populations. The interindividual variability in microbial composition is remarkable, but most individuals can be categorized into three different enterotypes, probably linked to long-term dietary habits [4]. One of the most important functional characteristics of the human microbiota is its diversity, i.e., species richness [2,5].

Commensal bacteria, and bacteria in the gut in particular, are essential for the development and maturation of the human immune system. Germ-free mice have significantly reduced lymph nodes in gut-associated lymphoid tissues [4]. Microbiota composition can affect immune cells in the gut via microbial components (LPS) or products of microbial metabolism (i.e., SCFA) [4]. Bacteroidetes and other Gram-negative bacteria contain lipopolysaccharides (LPS) in their cell wall, strong immune response activators [3]. Worsening of intestinal barrier function leads to leakage of gut bacteria components (termed microbe- or pathogen-associated molecular patterns—MAMPs or PAMPs) or even whole bacteria into the circulation. On the contrary, SCFA reduce pro-inflammatory cytokines production in monocytes and T cells and strengthen the tight junctions of intestinal epithelial cells, and butyrate-producing bacteria have beneficial immunometabolic effects [3]. These mechanisms may also explain the link between dysbiosis and autoimmune diseases [6,7].

## 3. Intestinal Barrier Function

The mucosal barrier is not only essential for the digestion and absorption of nutrients, but it also prevents the entry of diverse exterior antigens (food antigens, commensal bacteria, pathogens, and toxins). In the intestine, the front line of this barrier is an intact microbiome, followed by an intestinal mucus barrier and the most important part—a single layer of specialized epithelial cells linked together by tight junctions. Any alteration of the gut mucosal barrier increases the translocation of MAMPs into the lamina propria, triggering inflammatory cytokine-mediated changes. The “leaky gut” promotes both local and systemic immune responses. The gut barrier disruption creates the way for intestinal microbes and MAMPs to penetrate into the submucosa [8]. Non-occlusive intestinal ischemia during shock states, resulting in intestinal barrier disruption and bacterial translocation, has been associated with immune dysregulation, sepsis, and death in the critically ill [8]. The gut has been nicknamed a driver of multi-organ dysfunction in this patient population [9].

## 4. Changes in Gut Microbiota in Critically Ill Patients

Critical illness is an extreme alteration of homeostasis that requires medical and instrumental life support in addition to the treatment of the underlying disease. For the sake of the literature search for this review, we considered anyone treated in an intensive care unit as being critically ill. Specifically, we included patients with severe, fulminant, or complicated CDI [9,10,11] and active or fulminant IBD. As the human microbiome is a result of the complicated interplay between the host and the gut microbiota, it comes without surprise that critical illness is almost invariably associated with dysbiosis in a degree directly proportional with disease severity [10]. Most prominent is the relative increase in pathogenic bacteria (such as the Proteobacteria, Enterobacter and Staphylococcus) and a reduction in SCFA-producing protective microorganisms (such as Firmicutes and Bacteroidetes) and anti-inflammatory species as Faecalibacterium [11,12]. The dynamics of this microbiota alteration is astonishing. Ninety percent of the commensal organisms are lost within the first six hours of ICU stay [8]. Factors contributing to the dysbiosis of the critically ill can be summarized as follows:Artificial instrumentation of upper airways and upper GI tract (endotracheal intubation, nasogastric tube) overcomes natural immune barriers and leads to bacterial colonization of normally nearly sterile surfaces [11].Host responses to critical illness lead to ischemia-reperfusion injury of the gastrointestinal tract. This, in addition to the above discussed barrier disruption, also reduces the production of gastric protective mucus and the secretion of microbial peptides and IgA and reduces partial pressure of oxygen within and near intestinal wall [11].The lack of luminal nutrients in the gut causes catabolic starvation of bacteria, creating an additional selective pressure.The effects of medication have the potential to alter microbiota composition—for example, opioids reduce intestinal motility, and proton pump inhibitors alter the pH in the stomach. Nonetheless, by far the most disruptive factor is exposure to antibiotics. The US Centers for Disease Control found that 55% of all hospitalized patients received an antibiotic during their hospital stay. This proportion increased to 70% in the subgroup of patients in ICU [12,13]. One clinical manifestation of a profound microbiome alteration is antibiotic-associated diarrhea (AAD), which occurs in 5% to 35% of exposed subjects [12]. In addition, exposure to antibiotics increases *Clostridium difficile* (CD) or multidrug-resistant organisms (MDROs) colonization. Genes of antibiotic resistance then persist in the microbiome of the gut. This creates the rationale for the restoration of physiological microbiota by means of FMT, as discussed below.Environmental exposure to disinfectant agents and subtherapeutic concentrations of drugs likely plays a minor role, as healthy hospital workers do not seem to have significantly altered gut microbiota [14].

## 5. The Effect of Dysbiosis on Critically Ill Patients—Closing the Vicious Cycle

It is not only likely that the milieu in the human body affects microbiota but also that this relationship works in the opposite direction. Patients hospitalized with dysbiosis-associated diseases are at significantly increased risk of sepsis and septic shock [15]. Altered intestinal microbiota may lead to metabolic, immune, and even neurocognitive disturbances in the critically ill by one or more of the following mechanisms: Dysbiosis reduces fermentation of dietary fibers into SCFA—the main energy source for the colonic epithelium, which preserves gut integrity. In sepsis, there is an association between fecal butyrate concentration, pathogen translocation, and increased epithelial apoptosis [16]. Epithelial apoptosis results in diarrhea, malabsorption of nutrients, and fecal energy loss [10].Impaired intestinal barrier function leads to uncontrolled translocation of luminal contents into the body. The microbial products can cross the blood–brain barrier and contribute to the development of delirium and sepsis-associated encephalopathy [17].Dysbiosis reduces specific microbial stimulatory signals for T-helper cells and dysregulates the immune system, resulting in infectious complications [10]. These are made even more difficult to treat due to resistance genes preserved in the metagenome.Indeed, dysbiosis and MDRO colonization alters the bacterial ecology of ICUs and hospital floors, expanding its effect beyond the level of an individual patient.

In light of this, one of the possible ways of looking at the complicated interplay between the microbiome and the host is that dysbiosis of critical illness leads to a reduction in SCFA production, epithelial starvation, and damage, causing leaky gut with “spillage” of bacteria and MAMPs. This, in turn, increases systemic inflammation, further impairs gut barrier function and closing the vicious cycle. Indeed, all efforts in the ICU are put toward controlling underlying disease and supporting organ function, but can managing gut microbiota also be targeted?

## 6. Dysbiosis Therapy in ICU

The rich bidirectional relationship between the critically ill and their gut passengers (microbiota) is an attractive potential treatment target. Indeed, the very first step and probably the most important step in protecting gut microbiota is a strict antibiotic stewardship. Antibiotic overuse has repeatedly been associated with increased morbidity (including but not limited to *Clostridium difficile* infections [12]) and mortality [13] and with the emergence of MDROs [18]. Nonetheless, in many patients, antibiotic treatment is a necessary and lifesaving intervention. The question is then whether we can help patients to restore their damaged microbiome and whether such a restoration can improve patient-centered outcomes.

A large body of evidence from non-critical care settings is available on the use of prebiotics, probiotics, and fecal microbiota transplantation (FMT). Prebiotics are compounds in food that induce the growth or activity of beneficial microorganisms. Probiotics are living non-pathogenic microorganisms. The use of probiotics in critically ill patients may reduce the incidence of ventilator-associated pneumonia and antibiotic-associated diarrhea, but randomized controlled trials presented mixed results with regard to the influence on the length of ICU stay or mortality [19,20]. There were reports of severe sepsis caused by microorganisms contained in probiotic formulas, which were subsequently isolated from blood cultures [21]. Concerns arose in patients with severe acute pancreatitis, where enteral probiotics increased the rate of small bowel necrosis and death [22]. The apprehension to administer live bacteria into an upper gastrointestinal tract lined with altered epithelial barrier prevented probiotics from wider routine use in intensive care.

## 7. Fecal Microbial Transplantation: Principle and Use Outside of the Critical Care Setting

FMT is a procedure during which minimally processed feces from a healthy donor are transferred into a patient’s gut. Donor microbiota then engraft in the recipient and increase their microbiota diversity and restore normal bowel function in patients with dysbiosis-associated diseases such as *Clostridium difficile* infections (CDI), inflammatory bowel disease (IBD), irritable bowel syndrome (IBS), or metabolic syndrome [23]. In addition to living microorganisms, several other biologic products in the donor’s stool, such as bile acids, proteins, bacterial components, and bacteriophages, affect intestinal homeostasis after FMT. Most clinical evidence for FMT comes from studies in patients with CDI. Here, rather than treating the CDI with more antibiotics, restoring healthy microbiota which can “fight” with pathogens has been shown effective, in particular in recurrent CDI. Interestingly, one study demonstrated that the transfer of sterile stool filtrates also eliminated CDI symptoms, suggesting the importance of abiotic substance in the clinical effects of FMT [24].

Currently, FMT is recommended for recurrent CDI, with cure rate of about 90%, and as a rescue option in severe and fulminant CDI unresponsive to standard therapy in patients unfit for surgery. For the first episode of CDI, FMT is not yet an established treatment beyond the experimental setting [25,26,27].

A large amount of data from CDI patients allows us to make some assumptions with regard to safety and adverse effects of FMT. Even though FMT appears very safe, including in immune-compromised patients, there are risks associated with the application procedure, such as aspirations or gut perforation. The application into the lower gastrointestinal tract seems to have a better safety profile [28,29]. Indeed, although FMT is well-established in the treatment of CDI, there is no international consensus on the search for and testing of suitable donors, nor are there consistent international standard operating procedures for graft preparation [30].

## 8. Use of FMT in the Intensive Care Unit

(a) Severe forms of CDI. Critical illness can be a consequence of severe forms of CDI, for which FMT is a well-established treatment. In critically ill patients, CDI is responsible for 15–25% of nosocomial antibiotic-associated diarrhea. Not only are those patients at risk during their actual hospitalization but CDI also increases the risk of later readmission for sepsis by 70% [31,32]. The European CDI study (ECDIS) shows that 1 in 10 cases of CDI is either transferred to an intensive care unit, necessitates colectomy, or dies [29,33]. The population of critically ill patients is different, and data about the safety and efficacy of FMT from the studies in the general population cannot be directly transferable. In the ICU, patients are more vulnerable to developing CDI due to co-morbidities (DM, IBD, liver cirrhosis, CKD, malignancy), recent GI surgery [34,35,36,37], and higher exposure to exogenous risk factors such as antibiotics or other medications (immune suppression, PPI, H2 blocker, NSAID, laxatives) or invasive procedures (invasive mechanical ventilation, nasogastric intubation, prolonged use of laxatives). In addition, antibiotic stewardship is more challenging in the ICU setting, and administered antibiotics often have anti-anaerobic activity [37] or include clindamycin, cephalosporines, and/or fluoroquinolones. All of these are associated with increased risk of CDI [38]. The diagnosis of CDI could be challenging due to variety of other possible causes of diarrhea in ICU patients and the difficulty of detecting abdominal symptoms in sedated ventilated patients.

Indeed, despite the lack of high-quality evidence, first-line treatment for CDI in the critically ill is the same as in general population. Both vancomycin and metronidazole are active against the vegetative forms, but they are not sporicidal. Fidaxomicin, in addition to being active against vegetative forms, inhibits sporulation as well and has a narrower spectrum, thus less affecting the gut microbiota. Both of these factors translate to lower recurrence rate [2,39,40], but due to high cost, fidaxomicin remains the second line of treatment in most ICUs. In patients with fulminant colitis and/or septic shock refractory to conservative treatment, colectomy is recommended [37]. This invasive procedure bears a 50% mortality rate, which increases with age and severity of physiological deterioration [37]. Therefore, for the patients who do not have an absolute indication for surgery, such as colonic perforation, it would be beneficial to have an alternative treatment that would allow avoiding the surgery, mostly for the elderly and the sickest patients [31,33,41].

There is only one randomized controlled trial on FMT for fulminant or severe CDI in critically ill patients, and most of the data come from four retrospective case-cohort studies and uncontrolled studies (case reports and case series), as summarized in Table 1. Indeed, these data can be subjected to selection and publication biases and should be interpreted with great caution. Nonetheless, the available evidence suggests that FMT in critically ill ICU patients with recurrent, severe, or fulminant CDI is feasible and results in a reduction in mortality and morbidity compared with antibiotic therapy alone [42]. Importantly, there were only very few reported serious adverse events related to FMT.

Of note, rescue FMT was a promising alternative to colectomy in critically ill patients with severe and complicated CDI, with a primary cure rate of 78% (7/9), allowing 88% (8/9) of patients to avoid surgery [49]. There is a burning need for randomized controlled trials comparing standard of care and standard of care plus FMT in severe and fulminant forms of CDI.

(b) Critically ill patients with inflammatory bowel disease. IBD is an intestinal disorder that includes ulcerative colitis (UC) as well as Crohn’s disease (CD) and that is characterized by chronic inflammation of the gastrointestinal tract. A certain degree of dysbiosis is a hallmark of IBD and is associated with disease progression [67]. Microbes producing protective short chain fatty acids are reduced in IBD [68]. Patients with IBD are also at increased risk of developing CDI [69] and have worse outcomes, possibly due to IBD medication (repeated antibiotic courses, immunosuppression), altered immune and nutritional status, and frequent hospitalizations. Up to 20% of the cases of IBD flares tested positive for *Clostridium difficile*.

FMT for CDI patients who had underlying IBD had a lower success rate compared with patients without IBD, probably because of the severity of dysbiosis. Moreover, 26% of patients with IBD experienced a clinically significant flare of IBD immediately after FMT [70,71].

FMT has been attempted to improve microbial dysbiosis in IBD without CDI [72] and as a treatment of active IBD. There are RCTs showing a mild but statistically significant clinical, endoscopic, and histological improvement of active IBD in patients treated by FMT compared with placebo [68,73,74,75]. A proportion of these patients were critically ill (see Table 2). In addition, there is anecdotal evidence of a successful use of FMT as a rescue treatment to avoid surgery [76].

In light of this, FMT could be considered as a rescue treatment in critically ill patients before surgery in patients with refractory IBD. Most studies used rectal administration rather than upper GI, and the procedure appeared safe.

(c) FMT for septic shock and antibiotic-associated diarrhea. Several case reports and two case series on 31 patients described the use of FMT in septic shock with severe diarrhea in the ICU, mostly using the upper gastrointestinal tract as the way to deliver FMT. In these patients, FMT was intended to enrich the microbiome with commensals (mainly Firmicutes) and to reduce opportunistic organisms, and by doing so, to reduce systemic inflammation [28,90,91]. Repeated FMT was also used in the treatment of intestinal failure associated with drug-induced hypersensitivity syndrome [92] and severe antibiotic-associated diarrhea (AAD) [93]. The data are summarized in Table 3.

The uncontrolled nature of published studies does not allow to infer any conclusions about the effects of FMT in sepsis or for antibiotic-associated diarrhea, but it generates the hypothesis that FMT is safe and effective. Although there are also some experimental data supporting the use of FMT in sepsis [17], the biological plausibility seems much sounder for AAD, where FMT should be first tested in RCTs.

(d) Critically ill immunocompromised patients. Critically ill patients with immune suppression (HIV/AIDS, hematologic malignancies, or patients undergoing immune suppressive therapy for solid organ transplant or other reasons, etc.) represent a very specific subgroup, where introducing live microorganisms in the form of FMT could be most risky. Surprisingly, the immune-suppressed subgroup of patients with severe or fulminant CDI treated with FMT showed similarly high cure rates and no associated bacteremia or signs of worsened systemic inflammation [29,52]. FMT was also successfully used in three patients with severe refractory gastrointestinal acute graft-versus-host disease following allogeneic hematopoietic stem cell transplantation [98].

(e) FMT to eliminate colonization by multidrug-resistant organisms. Animal experiments showed that the restoration of the microbiome following FMT was associated with an immense reduction in the density of intestinal MDROs, probably by restricting their growth [99]. Indeed, critically ill patients exposed to broad spectrum antibiotics are often colonized with MDROs, and in theory, FMT could be a plausible alternative to selective bowel decontamination strategy by using antibiotics alone, offering an advantage of not threatening the bacterial ecology of intensive care units. An uncontrolled study of 20 immune-compromised hematologic patients demonstrated a total elimination of MDROs from the stool in 15 (75%) patients after FMT [100]. On the other hand, no effect of FMT was observed in an RCT. Thirty-nine immune competent patients colonized with MDROs were randomized to receive no treatment or a five-day course of nonabsorbable antibiotics followed by FMT. There was no significant difference in colonization rate in stool samples (MDRO eradication in 41% versus 29% in controls) [101]. Unfortunately, large scale RCTs measuring patient-centered and ecological outcomes are still missing.

## 9. Conclusions

FMT is an established treatment method for recurrent CDI, and this is also beneficial for patients who are critically ill or develop CDI as a consequence of IBD, immune deficiency, or protracted ICU stay. At the current level of evidence, FMT should be considered as a salvage treatment for the sickest patients with most severe forms of CDI in whom colectomy would otherwise be the only alternative. The biggest promise and most burning need of RCTs is in the treatment of post-antibiotic diarrhea, as FMT not only seems to eliminate symptoms but also may reduce the colonization rate of MDROs and improve systemic inflammation and outcomes. Current data suggest an acceptable safety profile of FMT administered into the lower gastrointestinal tract of critically ill patients, including those who are immune-suppressed, but due to the uncontrolled nature of most of the available trials, this warrants confirmation in large-scale randomized controlled trials.

## Figures and Tables

**Table 1 biomolecules-11-01459-t001:** Clinical studies on critically ill patients with *Clostridium difficile* infections.

Ref.	Study Type	Patients				Intervention (Fecal Microbial Transplantation)			Controls	Outcomes	
		N (FMT/Controls)//Critically Ill	Diagnosis	Age	Sex	Route of Administration	Frequency	Donor	No FMT Therapy	Beneficial	Adverse Events (Severe/Mild)
								R = Related, U = Unrelated, ? = Unknown; Fresh, Frozen, ? = Unknown			
[43]	Open label randomized clinical trial	56/0	sCDI	75	17M/39F	Lg-C	1 × 28 pt, multiple 28 pt	U (71%), R (29%)/mostly fresh	x	↓ AS, ↓ D	x/AS
[42]	Retrospective cohort study	225 (50 pt FMT)/205	fCDI, sCDI	61.2	123M/102F	Lg-C (98%)	median of 2 FMT	U/Fresh (10%) and Frozen	ATB therapy	↓ M	no comment
[33]		17/15	sCDI, cCDI	66,4	18M/14F	no comments	average 1.83 ± 0.7	?/?	no more details	↓ M	no comment
[44]		66/45	sCDI, scCDI	81 (69–87)	23M/43F	Ug-NGS	1 × 51 pt, 2 × 14 pt, 3 × 1 pt	R and U/Fresh (46%) and Frozen	Vanco p.o. +/– Metro i.v. or p.o. +/− FDX p.o.	↓ M	x/AS, F
[45]		16/32	sCDI, fCDI	62,6	7M/9F	Lg-C,S	every 3–5 days until resolution	R and U/?	Vanco p.o. +/− Metro i.v.	↓ M	1 × bacteremia (6.3%), 1 × perforation (6.3%)/no comment
[46]	Case series	14/0	sCDI, refCDI	73.4 (52–92)	5M/9F	Ug-Ngt (93%), Lg-C (7%)	1 × 10 pt, 2 × 2 pt, 3 × 2 pt	R (85.7 %) and U/Fresh.	not applicable	↓ AS, ↓ D	x/no comment
[47]		75/0	rCDI	76.4	21M/54F	Lg-C (88%), Lg-S (9.3%)	no comments	R (13.5%) and U/?		↓ AS, ↓ D	3 pt post-procedural hypotension, one case of perforation.
[48]		17/0	sCDI, cCDI	66.4 (38–89)	4M/13F	Lg-C (94%), E,S Ug-Njt	1 × 14 pt, 2 × 3 pt	R (58.8%) and U/?		↓ AS, ↓ D	x/AS
[49]		9/0	sCDI, cCDI	67.78	6M/3F	Ug-Njt (3×), Peg (1×) Lg-C (1×), Ug + Lg (C + Ngt) 4×	1 × 8 pt, 2 × 1 pt	U and R/?		↓ AS, ↓ D, ↓ Ilf, ↑ SA	x/no comment
[50]		328/0//42 pt sCDI	sCDI, rCDI	61.4 ± 19.3	87M/241F	Lg-C (76.9%)	no comments	?/?		no comment	no comment
[51]		64/0//26 pt sCDI	rCDI	74 (29–94)	25M/39F	Lg-C	1 × 44 pt, multifecal infusion 20 pt	R (44%) and U/Fresh (83%) and Frozen		↓ D	no comment
[52]		94/0	sCDI, fCDI + SOTp	56,3	47M/47F	Ug-Njt, Lg-C (81%), E (17%), S, Caps.	no details	R and U/Fresh (41%) and Frozen		↓ AS, ↓ D	3.2% severe diarrhea, AKI, fever, CMV reactivation/22,3% AS,D
[29]		80/0//36 pt sCDI, cCDI, refCDI	sCDI, refCD, recCDI + IC	53 (20–88)	42M/38F	Lg mostly	1 × 62 pt, no more comments	?/?		↓ AS, ↓ D	aspiration, mucosal tear caused by the colonoscopy/15% any SAE (AS, IBD flare.)
[45]		57/0	sCDI, scCDI	72 (60–79; 25–99)	23M/34F	Lg-C	1 × 30 pt, 2 × 16 pt, 3 × 4 pt, 4–5 × 2 pt	R and U/Fresh (51%) and Frozen		↓ AS, ↓ D, ↑ SA	x/no comment
[53]		146/0///s,cCDI 57(38.4%)	rCDI, sCDI, cCDI	78.6 (65 to 97)	46M/100F	Lg-C (80,8%), E,S Ug-Gfs, Ent	1 × 130 pt, multiple 16 pt	?/?		↓ AS, ↓ D	x/D, AS 11 pt (7,5%)
[54]		29/0	sCDI, scCDI	65,2 (25–92)	12M/17F	Lg-C	1 × 18 pt, 2 × 9 pt, 3 × 2 pt	R (36%) and U/?		↓ D, ↑ SA	x/no comment
[55]		35/0	sCDI	69 (29–91)	17M/18F	Lg-C	1 × 27 pt, multiple 8 pt	R (54%) and U/?		↓ AS, ↓ D, ↑ SA	no comment
[56]		4/0	sCDI	66–83	1M/3F	Lg-C	1 × 2 pt, 2 × 2 pt	U/Fresh (25%) and Frozen		↓ AS, ↓ D	no comment
[57]	Case report	1/0	sCDI	65	1M	Ug-Njt	1 × 1 pt	U/?		↓ AS, ↓ D	x/no comment
[58]		1/0	fCDI	69	1M	LG-E	1 × 1 pt	R/Fresh		↓ AS, ↓ D, ↓ Ilf	x/no comment
[59]		1/0	fCDI	26	1M	Lg-C	2 × 1 pt	R/Fresh		↓ AS, ↓ D, ↑ SA	x/no comment
[60]		1/0	sCDI	75	1F	Ug-Njt	1 × 1 pt	R/Fresh		↓ D, ↓ Ilf	x/no comment
[61]		1/0	sCDI, rCDI	65	1M	Lg-C	1 × 1 pt	R/Fresh		x	SIRS 4 days subsequent to the FMT without detecting an infectious cause
[62]		1/0	fCDI + AML	27	1M	Lg-S	1 × 1 pt	U/Frozen		↓ AS, ↓ D	x/no comment
[63]		1/0	CDI + HIV stage3	27	1M	Ug-Njt	1 × 1 pt	R/Fresh		↓ AS, ↓ D, ↓ Ilf	x/no comment
[38]		1/0	sCDI − liver Tx	47	1W	Ug-caps, Lg-S	2 × 1 pt	R/?		↓ AS, ↓ D, ↓ Ilf	x/no comment
[64]		1/0	sCDI + SCTx	21	1W	Ug-Njt	1 × 1 pt	R/?		↓ AS, ↓ D	x/no comment
[65]		1/0	fCDI + pBMcht	56	1M	Ug- Njt, Lg-C	11 × 1 pt (7 × C (days 2, 7, 8, 11, 12, 45, 48) + 4 × Njt (days 13, 14, 21, and 24)	U/Frozen		↓ AS, ↓ D	x/no comment
[66]		1/0	sCDI	71	1M	Lg-C	1×1 pt	R/Fresh		↓ AS, ↓ D	x/no comment

Note: x, none; pt, patient; D, diarrhea; F, fever; AS, abdominal symptoms; SS, septic symptoms; R, remission; M, mortality; Ilf, inflammatory laboratory findings; SA, surgery avoiding; CDI, Clostridium difficile infection; sCDI, severe CDI; scCDI, severe complicated CDI; rCDI- recurrent CDI; fCDI, fulminant CDI; Vanco, vancomycin; Metro, metronidazole; FDX—fidaxomicin. SEX: F, female; M, male. Route of administration: Ug, upper GI; Lg, lower GI; Ngt, nasogastric tube; Ngi, nasogastric infusion; Njt, nasojejunal tube; Ent, enteroscopy; C, colonoscopy; S, sigmoidoscopy; E, enema; Caps, capsule. IBD, inflammatory bowel disease: a, active; s, severe; ref, refractory; Tx, transplantation.

**Table 2 biomolecules-11-01459-t002:** Clinical studies on critically ill patients with inflammatory bowel diseases.

			Patient			Intervention (Fecal Microbial Transplantation)			Controls	Outcomes	
	Study Type	N (FMT /Controls)//Critically Ill	Diagnosis	Age	Sex	Route of Administration	Frequency	Donor	No FMT Therapy	Beneficial	Adverse Events (Severe/Mild)
								R = Related, U = Unrelated, ? = Unknown; Fresh, Frozen, ? = Unknown			
[73]	Double-blind placebo-controlled randomized trial	38/37	aUC	42.2 (FMT)/35.8 (placebo)	44M/31F	Lg-E	once weekly for 6 weeks	U/?	enema with placebo (water)	↑ R	1 pt in placebo gr urgent colectomy, 3 pt (1 pt in placebo gr 2 pt in FMT gr) rectal abscess, 1 pt in FMT gr CDI
[77]	Cohort study	17/19	mUC, sUC	40.4 y (FMT), 44.8 y (ATB)	13M/4F (FMT) 12M/7F (ATB)	Lg-C	1 × 17 pt	U and R/Fresh	ATB therapy	↓ AS	x/AS
[78]		17/10 pt//5 pt sUC	refUC	44 ± 18 (FMT), 36 ± 13 (ATB)	14M/3F (FMT), 3M/7F (ATB)	Lg-C + S	5× for 14 days (1 × C-4 × S)	U and R/Fresh	ATB therapy	↓ AS, ↑ R	no comment
[79]		55/37//52 pt (56%) extensive colitis	refUC, mUC, sUC	41.1 ± 13.9	56M/36F	Lg-C	1 × 55 pt	U and R/Fresh	ATB therapy	↓ AS	x/12 pt (13.0%) AS,D
[79]	Case series	30/0//20 pt (66.7 %) sIBD	refUC	34.6	14M/16F	Lg-C	1 × 27 pt, 2 × 3 pt	U (77%) and R/?	not applicable	↓ AS, ↓ D, ↑ R	x/AS 7 pt (23.3%)
[80]		14/0	refIBD (8 UC, 6 CD)	28–50 y	7M/7F	Ug-Njt (64%), Lg-C, E	2 × 5 pt, 4 × 9 pt (2 × Njt + 2 × C)	U (71%) and R/?		X (CD), ↑ R (UC)	1 pt aspiration pneumonia/4 pt high fever
[81]		14/0	refUC	47 ± 11	no details	Lg-C	1 × 5 pt, 2 × 1 pt, 4 × 3 pt, 6 × 2 pt	?/?		↓ AS	no comment
[82]		6/0	sUC, recUC	25–53	3M, 3F	Lg-E	daily for 5 days	U/?		↓ AS	no comment
[83]		9/0//6 pt (66%) sUC	mUC, sUC	47.90 (31–61)	7M, 2F	Lg-C (55.6%), Ug-Njt (44.4%)	3× (day 1, 3 and 5)	U/?		↓ AS, ↑ R	x/AS 33.3% (3/9)
[71]		30/0	refCD	38.0 ± 13.83	19M/11F	Ug-Njt	1 × 30 pt	U and R/?		↓ AS, ↑ R, ↑ BMI	x/F 2 pt
[84]		12/0//7 pt (58.3%) sUC	mUC, sUC	50.5 years (41–65)	M8/4F	Lg-C	multiple (no more comments)	U/?		↓ AS, ↑ R	x/x
[85]		67/0//15 pt (22.4%) sIBD	UC, CD + recCDI	45.42 ± 17.33	28M, 39F	Lg-C,S	1 × 60 pt, 2 × 6 pt, 3 × 1 pt	U/Fresh (88.1%)		↓ AS, ↓ D	x/AS
[86]		93/0	refUC, mUC, sUC	34.96 ± 11.27	58M/35F	Lg-C	7× (week 0, 2, 6, 10, 14, 18, 22)	U/Fresh		↓ AS	x/AS (30%)
[87]		10/0//7 pt (70%) sUC	aUC	31 (17–48)	7M/3F	Lg-C	1 × 10 pt	R/Fresh		x	x/6 pt exacerbation of the UC
[88]		16/0	aUC	37 (18–66)	10M, 6F	Ug-Gfs, Lg-C	3× for 2–3 months	U/?		↓ AS, ↓ Ilf, ↑ R	x/no comment
[89]	Case report	1/0	sUC	19	1M	Lg-C,E	3 × 1 pt	U/?		↓ AS, ↑ R	x/no comment

Note: x, none; pt, patient; D, diarrhea; F, fever; AS, abdominal symptoms; SS, septic symptoms; R, remission; M, mortality; Ilf, inflammatory laboratory findings; SA, surgery avoiding; CDI, *Clostridium difficile* infection; Vanco, vancomycin; Metro, metronidazole; FDX, fidaxomicin. SEX: F, female; M, male; Route of FMT administration: Ug, upper GI; Lg, lower GI; Ngt, nasogastric tube; Ngi, nasogastric infusion; Njt, nasojejunal tube; Ent, enteroscopy; C, colonoscopy; S, sigmoidoscopy; E, enema; Caps, capsule. IBD, inflammatory bowel disease: a, active; s, severe, ref, refractory; gr, group.

**Table 3 biomolecules-11-01459-t003:** Clinical studies on critically ill patients with sepsis and septic shock.

		Patient				Intervention			Controls	Outcomes	
	Study Type	N (FMT/Controls)//Critically Ill)	Diagnosis	Age	Sex	Route of Administration	Frequency	Donor	No FMT Therapy	Beneficial	Adverse Events (Severe/Mild)
								R = Related, U = Unrelated, ? = Unknown; Fresh, Frozen, ? = Unknown			
[93]	case series	18/0	Antibiotic-associated diarrhea, critical illness	55 (2–91)	12M/6F	Ug-Njt (13), Gfs (4), Lg-E (1)	1 × 8 pt, 2 × 7 pt, 3 × 1 pt, 4 × 2 pt	U/?	Not applicable	↓ SS, ↓ D, ↓ Ilf	x/7 pt (38.9%) FMT-related AEs (D,AS)
[90]	case report	1/0	Septic shock, watery diarrhea	44	1W	Ug-Njt	1 × 1 pt	R/Fresh		↓ SS, ↓ D	x/no comment
[91]		2/0	MODS, septic shock, severe diarrhea	65, 84	2M	Ug-Ngi	1 × 2 pt	U/?		↓ SS, ↓ D, ↓ F	x/no comment
[92]		1/0	MODS, drug-induced hypersensitivity syndrome	32	1F	Ug-Ngi	4×—every 6 days	U/?		↓ SS, ↓ D	x/no comment
[94]		1/0	Septic shock, severe diarrhea, UC	29	1F	Ug-Njt	1 × 1 pt	?/?		↓ SS, ↓ D, ↓ F, ↓ Ilf	x/no comment
[95]		1/0	MDRO infection, septic shock	57	1M	Ug-Peg	1 × 1 pt	?/?		------	the patient died the same day FMT was done
[28]		1/0	High-volume diarrhea (Apoptotic Enterocolitis) on ICU	16	1F	Lg-C	1 × 1 pt	R/?		↓ D	x/no comment
[96]		5/0	MRSA enteritis, septic shock	28 (19–45)	3M/2F	Ug-Njt	3×—once a day for 3 consecutive days	U and R/Fresh		↓ AS, ↓ D	x/no comment
[97]		1/0	MDRO Klebsiella, MODS	60	1M	Ug-Njt	2×—repeated after two weeks.	R/?		↓ SS, ↓ Ilf	x/no comment

Note: x, none; pt, patient; D, diarrhea; F, fever; AS, abdominal symptoms; SS, septic symptoms; R, remission; M, mortality; Ilf, inflammatory laboratory findings; SA, surgery avoiding; sCDI, severe CDI; scCDI, severe complicated CDI; rCDI, recurrent CDI; fCDI, fulminant CDI; Vanco, vancomycin; Metro, metronidazole; FDX, fidaxomicin. SEX: F, female; M, male. Routed of FMT administration: Ug, upper GI; Lg, lower GI; Ngt, nasogastric tube; Ngi, nasogastric infusion; Njt, nasojejunal tube; Ent, enteroscopy; C, colonoscopy; S, sigmoidoscopy; E, enema; Caps, capsule. IBD, inflammatory bowel disease: UC, ulcerative colitis; MODS, multiple organ dysfunction syndrome; MDRO, multidrug-resistant organism; MRSA, methicillin-resistant Staphylococcus aureus; ICU, intensive care unit.

## Data Availability

Data sharing not applicable. No new data were created or analyzed in this study. Data sharing is not applicable to this article.

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
