# Peer review of "Fecal Microbial Transplantation in Critically Ill Patients—Structured Review and Perspectives"

_biomolecules, 2021, doi:10.3390/biom11101459_

Round 1
Reviewer 1 Report
The current review "Faecal Microbial Transplantation in Critically Ill Patients – Structured Review and Perspectives" by Cibulková et al., explores to date knowledge on faecal microbiota transplantation (FMT) in the critically ill patients.
Overall the work is well structured and presented.
Minor comments
In the section "Changes in gut microbiota in critically ill patients" I would suggest a few lines describing which patients are considered as critically ill
Throughout the text there are some spelling errors and typos. Please revise your text
Author Response
Please see attached the response letter.

Reviewer 2 Report
The authors present a review concerning the interesting subject of microbiome modifications in critically ill patients. Despite the interesting topic and idea the information contained in this review is muddled. The information just touches the surface, is repetitive, unstructured and does not go into detail. The different fonts and styles of quoting throughout the manuscript are in line with this chaos.
In the following I will just give some details
Abstract:
“Because of splanchnic ischemia, exposure to antibiotics, and/or underlying the disease critically ill patients loose 90% of the commensal organisms in their gut within hours after the insult.”
REV: “…to antibiotics, and/or the underlying disease…”
Introduction:
REV: The information presented in the abstract is a little incoherent. You mention the epithelial barrier function and refer to the bacterial substrate responsible for cell vitality and barrier integrity later. I would suggest you only mention the key functions here and relate to the bacterial metabolites and their impact on the host in the later sections. In your introduction section (also in the following paragraphs you only superficially touch key functions of the microbiome. You repeat some informations (see comments below) but fail to give a clear and comprehensible overview of the key functions of the microbiome. A chain of events I would like to see is for instance: dysbiosis – lack of SCFA – leaky gut – “spillage” of MAMPs – increased inflammation – further increase of gut permeability. In a Journal with an IF of almost 5 you need to be more concise!
“It is estimated that the microbiota of a healthy human consists of between 500 and 2000 species
[1] (Rastelli et al., 2018).“
REV: Why do you use numbering and text for the reference here?
“Indeed, they also play important role in the host metabolism, converting dietary fiber to…”
REV: “…also play an important role…”
Intestinal microbiota diversity…:
“The interindividual variability in microbial composition is remarkable, but most individuals can be categorized into three different enterotypes, probably linked to long-term dietary habits (Wen & Duffy, 2017).”
REV: Either you use numbering or text for the references – refer to the author guidelines and stick to either throughout the manuscript please.
“Dysbiosis, a state with low bacterial diversity, in which the homeostasis of the gut microbiome is disrupted, has been associated with a range of diseases [1,3].”
REV: You have mentioned this above
“…of gut epithelia cells…”
REV: …intestinal epithelial cells…
Intestinal barrier function:
“In the intestine, the front line of this barrier is only a single layer of specialized epithelial cells that are linked together by tight junctions.”
REV: This remark requires literature. Furthermore, I disagree: the front line is an intact microbiome followed by the intestinal mucus barrier and the IECs and the interconnecting tight junctions. I agree that the IECs are the most important component of the barrier.
Changes in gut microbiota in critically ill patients:
REV: Again, you mangle the way you quote references here. Additionally, you switch between different fonts. If you should at least adapt the font.
Author Response

(The authors gave the same response as above.)
